# Do We Need Anisotropic Graph Neural Networks?

**Shyam A. Tailor**[*1]    **Felix L. Opolka**[1,2]    **Pietro Liò**[1]    **Nicholas D. Lane**[1,3]

[1]Department of Computer Science and Technology, University of Cambridge
[2]Invenia Labs, Cambridge, UK
[3]Samsung AI Center, Cambridge, UK

## Abstract

Common wisdom in the graph neural network (GNN) community dictates that *anisotropic* models—in which messages sent between nodes are a function of both the source and target node—are required to achieve state-of-the-art performance. Benchmarks to date have demonstrated that these models perform better than comparable *isotropic* models—where messages are a function of the source node only. In this work we provide empirical evidence challenging this narrative: we propose an isotropic GNN, which we call Efficient Graph Convolution (EGC), that consistently outperforms comparable anisotropic models, including the popular GAT or PNA architectures by using spatially-varying adaptive filters. In addition to raising important questions for the GNN community, our work has significant real-world implications for efficiency. EGC achieves higher model accuracy, with lower memory consumption and latency, along with characteristics suited to accelerator implementation, while being a drop-in replacement for existing architectures. As an isotropic model, it requires memory proportional to the number of vertices in the graph ($\mathcal{O}(V)$); in contrast, anisotropic models require memory proportional to the number of edges ($\mathcal{O}(E)$). We demonstrate that EGC outperforms existing approaches across 6 large and diverse benchmark datasets, and conclude by discussing questions that our work raise for the community going forward. Code and pretrained models for our experiments are provided at https://github.com/shyam196/egc.

## 1 Introduction

Graph Neural Networks (GNNs) have emerged as an effective way to build models over arbitrarily structured data. For example, they have successfully been applied to computer vision tasks: GNNs can deliver high performance on point cloud data (Qi et al., 2017) and for feature matching across images (Sarlin et al., 2020). Recent work has also shown that they can be applied to physical simulations (Pfaff et al., 2020; Sanchez-Gonzalez et al., 2020). Code analysis is another application domain where GNNs have found success (Guo et al., 2020; Allamanis et al., 2017).

In recent years, the research community has devoted significant attention to building more expressive, and better performing, models to process graphs. Efforts to benchmark GNN models, such as Open Graph Benchmark (Hu et al., 2020), or the work by Dwivedi et al. (2020), have attempted to more rigorously quantify the relative performance of different proposed architectures. One common conclusion—explicitly stated by Dwivedi et al. (2020)—is that *anisotropic*[1] models, in which messages sent between nodes are a function of both the source and target node, are the best performing models. By comparison, isotropic models, where messages are a function of the source node only, achieve lower accuracy, even if they have efficiency benefits over comparable anisotropic models. Intuitively, this conclusion is satisfying: anisotropic models are inherently more expressive, hence we would expect them to perform better in most situations. Our work provides a surprising challenge to this wisdom by providing an isotropic model, called Efficient Graph Convolution (**EGC**),

---

[*]Corresponding author. Contact at sat62@cam.ac.uk
[1]Definition from Dwivedi et al. (2020). Equal to attentional & message passing from Bronstein et al. (2021).

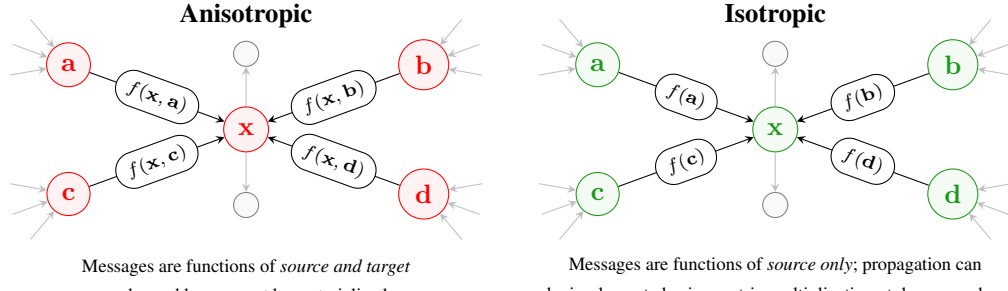

Figure 1: Many GNN architectures (e.g. GAT (Veličković et al., 2018), PNA (Corso et al., 2020)) incorporate sophisticated message functions to improve accuracy (left). This is problematic as we must materialize messages, leading to $\mathcal{O}(E)$ memory consumption and OPs to calculate messages; these dataflow patterns are also difficult to optimize for at the hardware level. **This work demonstrates that we can use simple message functions, requiring only $\mathcal{O}(V)$ memory consumption** (right) **and improve performance over existing GNNs.**

that outperforms comparable anisotropic approaches, including the popular GAT (Veličković et al., 2018) and PNA (Corso et al., 2020) architectures.

In addition to providing a surprising empirical result for the community, our work has significant practical implications for efficiency, as shown in Figure 1. As EGC is an isotropic model achieving high accuracy, we can take advantage of the efficiency benefits offered by isotropic models without having to compromise on model accuracy. We have seen memory consumption and latency for state-of-the-art GNN architectures increase to $\mathcal{O}(E)$ in recent years, due to state-of-the-art models incorporating anisotropic mechanisms to boost accuracy. EGC reduces the complexity to $\mathcal{O}(V)$, delivering substantial real-world benefits, albeit with the precise benefit being dependent on the topology of the graphs the model is applied to. The reader should note that our approach can also be combined with other approaches for improving the efficiency of GNNs. For example, common hardware-software co-design techniques include quantization and pruning (Sze et al., 2020) could be combined with this work, which proposes an orthogonal approach of improving model efficiency by improving the underlying architecture design. We also note that our approach can be combined with graph sampling techniques (Zeng et al., 2019; Hamilton et al., 2017; Chen et al., 2018a) to improve scalability further when training on graphs with millions, or billions, of nodes.

**Contributions** **(1)** We propose a new GNN architecture, Efficient Graph Convolution (*EGC*), and provide both spatial and spectral interpretations for it. **(2)** We provide a rigorous evaluation of our architecture across 6 large graph datasets covering both transductive and inductive use-cases, and demonstrate that EGC consistently achieves better results than strong baselines. **(3)** We provide several ablation studies to motivate the selection of the hyperparameters in our model. **(4)** We demonstrate that our model simultaneously achieves better parameter efficiency, latency and memory consumption than competing approaches. Code and pre-trained models for our experiments (including baselines) can be found at `https://github.com/shyam196/egc`. At time of publication, EGC has also been upstreamed to PyTorch Geometric (Fey & Lenssen, 2019).

## 2 BACKGROUND

### 2.1 HARDWARE-SOFTWARE CO-DESIGN FOR DEEP LEARNING

Several of the popular approaches for co-design have already been described in the introduction: quantization, pruning, and careful architecture design are all common for CNNs and Transformers (Vaswani et al., 2017). In addition to enabling better performance to be obtained from general purpose processors such as CPUs and GPUs, these techniques are also essential for maximizing the return from specialized accelerators; while it may be possible to improve performance over time due to improvements in CMOS technology, further improvements plateau without innovation at the algorithmic level (Fuchs & Wentzlaff, 2019). As neural network architecture designers, we cannot simply rely on improvements in hardware to make our proposals viable for real-world deployment.

## 2.2 GRAPH NEURAL NETWORKS

Many GNN architectures can be viewed as a generalization of CNN architectures to the irregular domain: as in CNNs, representations at each node are built based on the local neighborhood using parameters that are shared across the graph. GNNs differ as we cannot make assumptions about the the size of the neighborhood, or the ordering. One common framework used to define GNNs is the message passing neural network (MPNN) paradigm (Gilmer et al., 2017). A graph $\mathcal{G} = (V, E)$ has node features $\mathbf{X} \in \mathbb{R}^{N \times F}$, adjacency matrix $\mathbf{A} \in \mathbb{R}^{N \times N}$ and optionally $D$-dimensional edge features $\mathbf{E} \in \mathbb{R}^{E \times D}$. We define a function $\phi$ that calculates messages from node $u$ to node $v$, a differentiable and permutation-invariant aggregator $\oplus$, and an update function $\gamma$ to calculate representations at layer $l + 1$: $\mathbf{h}_{l+1}^{(i)} = \gamma(\mathbf{h}_l^{(i)}, \oplus_{j \in \mathcal{N}(i)}[\phi(\mathbf{h}_l^{(i)}, \mathbf{h}_l^{(j)}, \mathbf{e}_{ij})])$. Propagation rules for baseline architecture are provided in Table 1, with further details supplied in Table 5 in the Appendix.

**Relative Expressivity of GNNs**    Common wisdom in the research community states that isotropic GNNs are less expressive than anisotropic GNNs; empirically this is well supported by benchmarks. Brody et al. (2022) prove that GAT models can be strictly more expressive than isotropic models. Bronstein et al. (2021) also discuss the relative expressivity of different classes of GNN layer, and argue that convolutional (also known as isotropic) models are well suited to problems leveraging homophily[2] in the input graph. They further argue that attentional, or full message passing, models are suited to handling heterophilous problems, but they acknowledge the resource consumption and trainability of these architectures may be prohibitive—especially in the case of full message passing.

**Scaling and Deploying GNNs**    While GNNs have seen success across a range of domains, there remain challenges associated with scaling and deploying them. Graph sampling is one approach to scaling training for large graphs or models which will not fit in memory. Rather than training over the full graph, each iteration is run over a sampled sub-graph; approaches vary in whether they sample node-wise (Hamilton et al., 2017), layer-wise (Chen et al., 2018a; Huang et al., 2018), or sub-graphs (Zeng et al., 2019; Chiang et al., 2019). Alternatively, systems for distributed GNN training have been proposed (Jia et al., 2020) to scale training beyond the limits of a single accelerator. Some works have proposed architectures that are designed to accommodate scaling: graph-augmented MLPs, such as SIGN (Rossi et al., 2020), are explicitly designed as a shallow architecture, as all the graph operations are done as a pre-processing step. Other work includes applying neural architecture search (NAS) to arrange existing GNN layers (Zhao et al., 2020), or building quantization techniques for GNNs (Tailor et al., 2021). Finally, a recent work has shown that using memory-efficient reversible residuals (Gomez et al., 2017) for GNNs (Li et al., 2021) enables us to train far deeper and larger GNN models than before, thereby progressing the state-of-the-art accuracy.

**Why Are Existing Approaches Not Sufficient?**    It is worth noting that many of these approaches have significant limitations that we aim to address with our work. Sampling methods are often ineffective when applied to many problems which involve model generalization to unseen graphs—a common use-case for GNNs. We evaluated a variety of sampling approaches and observed that even modest sampling levels, which provide little benefit to memory or latency, cause model performance to decline noticeably. In addition, these methods do not accelerate the underlying GNN, hence they may not provide any overall benefit to *inference* latency. There is also no evidence that we are aware of that graph-augmented MLPs perform adequately when generalizing to unseen graphs; indeed, they are known to be theoretically less expressive than standard GNNs (Chen et al., 2021). We also investigated this setup, and found that these approaches do not offer competitive accuracy with state-of-the-art approaches. Experiment details and results, along with further discussion of the limitations of existing work, is provided in Appendix B.

In summary, our work on efficient GNN architecture design is of interest to the community for two reasons: firstly, it raises questions about common assumptions, and how we design and evaluate GNN models; secondly, our work may enable us to scale our models further, potentially yielding improvements in accuracy. In addition, for tasks where we need to generalize to unseen graphs, such as code analysis or point cloud processing, we reduce memory consumption and latency, thereby enabling us to deploy our models to more resource-constrained devices than before. We note that efficient architecture design can be usefully combined with other approaches including sampling, quantization, and pruning, where appropriate.

---

[2]Homophily means that if two nodes are connected, then they have high similarity

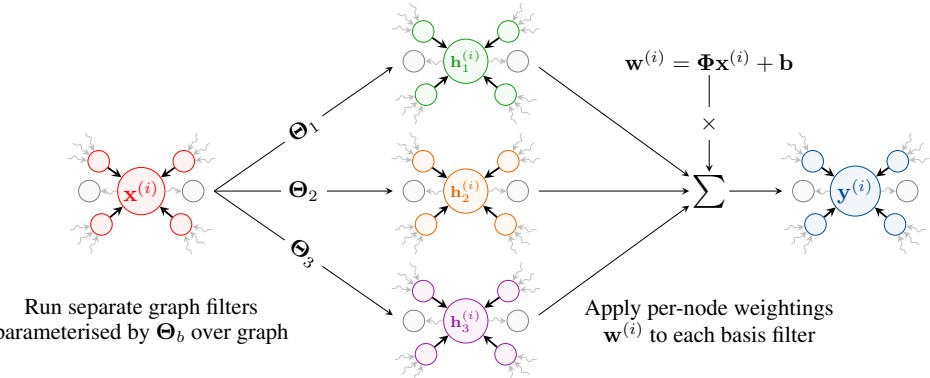

Figure 2: Visual representation of our EGC-S layer. In this visualization we have 3 basis filters (i.e. $B = 3$), which are combined using per-node weightings $\mathbf{w}$. This simplified figure does not show the usage of heads, or multiple aggregators, as used by EGC-M.

# 3 OUR ARCHITECTURE: EFFICIENT GRAPH CONVOLUTION (EGC)

In this section we describe our approach, and delay theoretical analysis to the next section. We present two versions: *EGC-S*(ingle), using a single aggregator, and *EGC-M*(ulti) which generalizes our approach by incorporating multiple aggregators. Our approach is visualized in Figure 2.

## 3.1 ARCHITECTURE DESCRIPTION

For a layer with in-dimension of $F$ and out-dimension of $F'$ we use $B$ basis weights $\mathbf{\Theta}_b \in \mathbb{R}^{F' \times F}$. We compute the output for node $i$ by calculating combination weighting coefficients $\mathbf{w}^{(i)} \in \mathbb{R}^B$ *per node*, and weighting the results of each aggregation using the different basis weights $\mathbf{\Theta}_b$. The output for node $i$ is computed in three steps. First, we perform the aggregation with each set of basis weights $\mathbf{\Theta}_b$. Second, we compute the weighting coefficients $\mathbf{w}^{(i)} = \mathbf{\Phi}\mathbf{x}^{(i)} + \mathbf{b} \in \mathbb{R}^B$ for each node $i$, where $\mathbf{\Phi} \in \mathbb{R}^{B \times F}$ and $\mathbf{b} \in \mathbb{R}^B$ are weight and bias parameters for calculating the combination weighting coefficients. Third, the layer output for node $i$ is the weighted combination of aggregation outputs:

$$\mathbf{y}^{(i)} = \sum_{b=1}^{B} w_b^{(i)} \sum_{j \in \mathcal{N}(i)} \alpha(i,j) \mathbf{\Theta}_b \mathbf{x}^{(j)} \tag{1}$$

where $\alpha(i,j)$ is some function of nodes $i$ and $j$, and $\mathcal{N}(i)$ denotes the in-neighbours of $i$. A popular method pioneered by GAT (Veličković et al., 2018) to boost representational power is to represent $\alpha$ using a learned function of the two nodes' representations. While this enables anisotropic treatment of neighbors, and can boost performance, it necessarily results in memory consumption of $\mathcal{O}(E)$ due to messages needing to be explicitly materialized, and complicates hardware implementation for accelerators. If we choose a representation for $\alpha$ that is not a function of the node representations—such as $\alpha(i,j) = 1$ to recover the add aggregator used by GIN (Xu et al., 2019), or $\alpha(i,j) = 1/\sqrt{\deg(i)\deg(j)}$ to recover symmetric normalization used by GCN (Kipf & Welling, 2017)—then we can implement our message propagation phase using sparse matrix multiplication (SpMM), and avoid explicitly materializing each message, even for the backwards pass. In this work, we assume $\alpha(i,j)$ to be symmetric normalization as used by GCN unless otherwise stated; we use this normalization as it is known to offer strong results across a variety of tasks; more formal justification is provided in section 4.2.

**Adding Heads as a Regularizer** We can extend our layer through the addition of heads, as used in architectures such as GAT or Transformers (Vaswani et al., 2017). These heads share the basis weights, but apply different weighting coefficients per head. We find that adding this degree of freedom aids regularization when the number of heads ($H$) is larger than $B$, as bases are discouraged from specializing (see section 5.3), without requiring the integration of additional loss terms into the optimization—hence requiring no changes to code for downstream users. To normalize the output

dimension, we change the basis weight matrices dimensions to $\frac{F'}{H} \times F$. Using $\|$ as the concatenation operator, and making the use of symmetric normalization explicit, we obtain the **EGC-S** layer:

$$\mathbf{y}^{(i)} = \bigg\|_{h=1}^{H} \sum_{b=1}^{B} w_{h,b}^{(i)} \sum_{j \in \mathcal{N}(i) \cup \{i\}} \frac{1}{\sqrt{\deg(i)\deg(j)}} \mathbf{\Theta}_b \mathbf{x}^{(j)} \qquad (2)$$

EGC works by combining basis matrices. This idea was proposed in R-GCN (Schlichtkrull et al., 2018) to handle multiple edge types; Xu et al. (2021) can be viewed as a generalization of this approach to point cloud analysis. In this work we are solving a different problem to these works: we are interested in designing efficient architectures, rather than new ways to handle edge information.

## 3.2 BOOSTING REPRESENTATIONAL CAPACITY

Recent work by Corso et al. (2020) has shown that using only a single aggregator is sub-optimal: instead, it is better to combine several different aggregators. In Equation (2) we defined our layer to use only symmetric normalization. To improve performance, we propose applying different aggregators to the representations calculated by $\mathbf{\Theta}_b \mathbf{x}^{(j)}$. The choice of aggregators could include different variants of summation aggregators e.g. mean or unweighted addition, as opposed to symmetric normalization that was proposed in the previous section. Alternatively, we can use aggregators such as stddev, min or max which are not based on summation. It is also possible to use directional aggregators proposed by Beaini et al. (2021), however this enhancement is orthogonal to this work. If we have a set of aggregators $\mathcal{A}$, we can extend Equation (2) to obtain our **EGC-M** layer:

$$\mathbf{y}^{(i)} = \bigg\|_{h=1}^{H} \sum_{\oplus \in \mathcal{A}} \sum_{b=1}^{B} w_{h,\oplus,b}^{(i)} \bigoplus_{j \in \mathcal{N}(i) \cup \{i\}} \mathbf{\Theta}_b \mathbf{x}^{(j)} \qquad (3)$$

where $\oplus$ is an aggregator. With this formulation, we are reusing the same messages we have calculated as before—but we are applying several aggregation functions to them at the same time.

**Aggregator Fusion** It would appear that adding more aggregators would cause latency and memory consumption to grow linearly. However, this is not true in practice. Firstly, since sparse operations are typically memory bound in practice, we can apply extra aggregators to data that has already arrived from memory with little latency penalty. EGC can also efficiently inline the node-wise weighting operation at inference time, thereby resulting in relatively little memory consumption overhead. The equivalent optimization is more difficult to apply successfully to PNA due to the larger number of operations per-node that must be performed during aggregation, caused by scaling functions being applied to each aggregation, before all results are concatenated and a transformation applied. More details, including profiling and latency measurements, can be found in Appendix D.

# 4 INTERPRETATION AND BENEFITS

This section will explain our design choices, and why they are better suited to the hardware. We emphasize that our approach does *not* directly correspond to attention.

## 4.1 SPATIAL INTERPRETATION: NODE-WISE WEIGHT MATRICES

In our approach, each node effectively has its own weight matrix. We can derive this by re-arranging eq. (2) by factorizing the $\mathbf{\Theta}_b$ terms out of inner sum:

$$\mathbf{y}^{(i)} = \bigg\|_{h=1}^{H} \underbrace{\mathbf{\Theta}_h^{(i)}}_{\text{Varying per Node}} \underbrace{\left( \sum_{j \in \mathcal{N}(i) \cup \{i\}} \frac{1}{\sqrt{\deg(i)\deg(j)}} \mathbf{x}^{(j)} \right)}_{\text{Computable via SpMM}} \qquad (4)$$

In contrast, GAT shares weights, and pushes complexity into the message calculation phase by calculating per-message weightings. MPNN (Gilmer et al., 2017) and PNA (Corso et al., 2020) further increase complexity by explicitly calculating each message—resulting in substantial latency overhead due to the number of dense operations increasing by roughly $\frac{|E|}{|V|}$. Specifically, we have:

$$\mathbf{y}_{\text{GAT}}^{(i)} = \Big\|_{h=1}^{H} \underbrace{\Theta}_{\text{Shared Weights}} \underbrace{\left(\sum_{j \in \mathcal{N}(i) \cup \{i\}} \alpha_{h,i,j} \mathbf{x}^{(j)}\right)}_{\text{Calculated Message Weighting}} \qquad \mathbf{y}_{\text{PNA}}^{(i)} = U(\mathbf{x}^{(i)}, \bigoplus_{j \in \mathcal{N}(i)} \underbrace{M(\mathbf{x}^{(i)}, \mathbf{x}^{(j)})}_{\substack{\text{Explicit Message} \\ \text{Calculation}}})$$

From an efficiency perspective we observe that our approach of using SpMM has better characteristics due to it requiring only $\mathcal{O}(V)$ memory consumption—no messages must be explicitly materialized to use SpMM. We note that although it is possible to propagate messages for GAT with SpMM, there is no way to avoid storing the weightings during training as they are needed for backpropagation, resulting in $\mathcal{O}(E)$ memory consumption. We also note that fusing the message and aggregation steps for certain architectures may be possible at inference time, but this is a difficult pattern for hardware accelerators to optimize for.

**Relation To Attention**    Our method is *not directly related to attention*, which relies upon pairwise similarity mechanisms, and hence results in a $\mathcal{O}(E)$ cost when using the common formulations. Alternatives to attention-based Transformers proposed by Wu et al. (2019a) are a closer analogue to our technique, but rely upon explicit prediction of the per-timestep weight matrix. This approach is not viable for graphs, as the neighborhood size is not constant.

## 4.2 Spectral Interpretation: Localised Spectral Filtering

We can also interpret our EGC-S layer through the lens of graph signal processing (Sandryhaila & Moura, 2013). Many modern graph neural networks build on the observation that the convolution operation for the Euclidean domain when generalised to the graph domain has strong inductive biases: it respects the structure of the domain and preserves the locality of features by being an operation localised in space. Our method can be viewed as a method of building adaptive filters for the graph domain. Adaptive filters are a common approach when signal or noise characteristics vary with time or space; for example, they are commonly applied in adaptive noise cancellation. Our approach can be viewed as constructing adaptive filters by linearly combining learnable filter banks with spatially varying coefficients.

The graph convolution operation is typically defined on the spectral domain as filtering the input signal $\mathbf{x} \in \mathbb{R}^N$ on a graph with $N$ nodes with a filter $g_\theta$ parameterized by $\theta$. This requires translating between the spectral and spatial domain using the Fourier transform. As on the Euclidean domain, the Fourier transform on the graph-domain is defined as the basis decomposition with the orthogonal eigenbasis of the Laplace operator, which for a graph with adjacency matrix $\mathbf{A} \in \mathbb{R}^{N \times N}$ is defined as $\mathbf{L} = \mathbf{D} - \mathbf{A}$, where $\mathbf{D}$ is the diagonal degree matrix with $D_{ii} = \sum_{j=1}^{N} A_{ij}$. The Fourier transform of a signal $\mathbf{x} \in \mathbb{R}^N$ then is $\mathcal{F}(\mathbf{x}) = \mathbf{U}^\top \mathbf{x}$, where $\mathbf{L} = \mathbf{U} \mathbf{\Lambda} \mathbf{U}^\top$, with orthogonal eigenvector-matrix $\mathbf{U} \in \mathbb{R}^{N \times N}$ and diagonal eigenvalue-matrix $\mathbf{\Lambda} \in \mathbb{R}^{N \times N}$. The result of a signal $\mathbf{x}$ filtered by $g_\theta$ is $\mathbf{y} = g_\theta(\mathbf{L})\mathbf{x} = \mathbf{U} g_\theta(\mathbf{\Lambda}) \mathbf{U}^\top \mathbf{x}$ where the second equality holds if the Taylor expansion of $g_\theta$ exists.

Our approach corresponds to learning multiple filters and computing a linear combination of the resulting filters with weights depending on the attributes of each node locally. The model therefore allows applying multiple filters for each node, enabling us to obtain a spatially-varying frequency response, while staying far below $\mathcal{O}(E)$ in computational complexity. Using a linear combination of filters, the filtered signal becomes $\mathbf{y} = \sum_{b=1}^{B} \mathbf{w}_b \odot g_{\theta_b}(\mathbf{L})\mathbf{x}$, where $\mathbf{w}_b \in \mathbb{R}^N$ are the weights of filter $b$ for each of the $N$ nodes in the graph. If we parameterize our filter using first-order Chebyshev polynomials as used by Kipf & Welling (2017) our final expression for the filtered signal becomes $\mathbf{Y} = \sum_{b=1}^{B} \mathbf{w}_b \odot (\tilde{\mathbf{D}}^{-\frac{1}{2}} \tilde{\mathbf{A}} \tilde{\mathbf{D}}^{-\frac{1}{2}}) \mathbf{X} \mathbf{\Theta}_b$, where $\tilde{\mathbf{A}} = \mathbf{A} + \mathbf{I}_N$ is the adjacency matrix with added self-loops and $\tilde{\mathbf{D}}$ is the diagonal degree matrix of $\tilde{\mathbf{A}}$ as defined earlier. This justifies the symmetric normalization aggregator we chose in Equation (2).

Cheng et al. (2021) proposed an approach for localized filtering. However, their approach does not generalize to unseen topologies or scale to large graphs as it requires learning the coefficients of several filter matrices $\mathbf{S}_k$ of size $N \times N$. Our approach does not suffer from these constraints.

| Architecture | Propagation Rule | Memory | ZINC (MAE ↓) Unseen Graph Regression | CIFAR (Acc. ↑) Unseen Graph Classification | MolHIV (ROC-AUC ↑) Unseen Graph Classification | Code-V2 (F1 ↑) Unseen Graph Classification | Arxiv (Acc. ↑) Transductive Node Classification |
|---|---|---|---|---|---|---|---|
| GCN | $\mathbf{y}^{(i)} = \Theta \sum_j \frac{1}{\sqrt{\deg(i)\deg(j)}} \mathbf{x}^{(j)}$ | $\mathcal{O}(V)$ | $0.459 \pm 0.006$ | $55.71 \pm 0.38$ | $76.14 \pm 1.29$ | $0.1480 \pm 0.0018$ | $71.92 \pm 0.21$ |
| GIN | $\mathbf{y}^{(i)} = f_\Theta[(1 + \epsilon)\mathbf{x}^{(i)} + \sum_j \mathbf{x}^{(j)}]$ | $\mathcal{O}(V)$ | $0.387 \pm 0.015$ | $55.26 \pm 1.53$ | $76.02 \pm 1.35$ | $0.1481 \pm 0.0027$ | $67.33 \pm 1.47$ |
| GraphSAGE | $\mathbf{y}^{(i)} = \Theta_1 \mathbf{x}^{(i)} + \bigoplus_j \Theta_2 \mathbf{x}^{(j)}$ | $\mathcal{O}(V)$ | $0.468 \pm 0.003$ | $65.77 \pm 0.31$ | $75.97 \pm 1.69$ | $0.1453 \pm 0.0028$ | $71.73 \pm 0.26$ |
| GAT | $\mathbf{y}^{(i)} = \alpha_{i,i}\Theta\mathbf{x}^{(i)} + \sum_j \alpha_{i,j}\Theta\mathbf{x}^{(j)}$ | $\mathcal{O}(E)$ | $0.475 \pm 0.007$ | $64.22 \pm 0.46$ | $77.17 \pm 1.37$ | $0.1513 \pm 0.0011$ | $* 71.81 \pm 0.23$ |
| GATv2 | $\mathbf{y}^{(i)} = \alpha_{i,i}\Theta\mathbf{x}^{(i)} + \sum_j \alpha_{i,j}\Theta\mathbf{x}^{(j)}$ | $\mathcal{O}(E)$ | $0.447 \pm 0.015$ | $67.48 \pm 0.53$ | $77.15 \pm 1.55$ | $0.1537 \pm 0.0022$ | $* 71.87 \pm 0.43$ |
| MPNN-Sum | $\mathbf{y}^{(i)} = U(\mathbf{x}^{(i)}, \sum_j M(\mathbf{x}^{(i)}, \mathbf{x}^{(j)}))$ | $\mathcal{O}(E)$ | $0.381 \pm 0.005$ | $65.39 \pm 0.47$ | $75.19 \pm 3.57$ | $0.1470 \pm 0.0017$ | $* 66.11 \pm 0.56$ |
| MPNN-Max | $\mathbf{y}^{(i)} = U(\mathbf{x}^{(i)}, \max_j M(\mathbf{x}^{(i)}, \mathbf{x}^{(j)}))$ | $\mathcal{O}(E)$ | $0.468 \pm 0.002$ | $69.70 \pm 0.55$ | $77.07 \pm 1.37$ | $0.1552 \pm 0.0022$ | $* 71.02 \pm 0.21$ |
| PNA | $\mathbf{y}^{(i)} = U(\mathbf{x}^{(i)}, \bigoplus_j M(\mathbf{x}^{(i)}, \mathbf{x}^{(j)}))$ | $\mathcal{O}(E)$ | $0.320 \pm 0.032$ | $70.21 \pm 0.15$ | $\mathbf{79.05 \pm 1.32}$ | $* 0.1570 \pm 0.0032$ | $* 71.21 \pm 0.30$ |
| **EGC-S** *(Ours)* | Equation 2 | $\mathcal{O}(V)$ | $0.364 \pm 0.020$ | $66.92 \pm 0.37$ | $77.44 \pm 1.08$ | $0.1528 \pm 0.0025$ | $\mathbf{72.21 \pm 0.17}$ |
| **EGC-M** *(Ours)* | Equation 3 | $\mathcal{O}(V)$ | $\mathbf{0.281 \pm 0.007}$ | $\mathbf{71.03 \pm 0.42}$ | $78.18 \pm 1.53$ | $\mathbf{0.1595 \pm 0.0019}$ | $71.96 \pm 0.23$ |

Table 1: Results (mean ± standard deviation) for *parameter-normalized* models run on 5 datasets. Details of the specific aggregators chosen per dataset and further experimental details can be found in the supplementary material. Results marked with ∗ ran out of memory on 11GB 1080Ti and 2080Ti GPUs. EGC obtains best performance on 4 of the tasks, with consistently wide margins.

# 5 EVALUATION

## 5.1 PROTOCOL

We primarily evaluate our approach on 5 datasets taken from recent works on GNN benchmarking. We use ZINC and CIFAR-10 Superpixels from Dwivedi et al. (2020) and Arxiv, MolHIV and Code from Open Graph Benchmark (Hu et al., 2020). These datasets cover a wide range of domains, cover both transductive and inductive tasks, and are larger than datasets which are typically used in GNN works. We use evaluation metrics and splits specified by these papers. Baseline architectures chosen reflect popular general-purpose choices (Kipf & Welling, 2017; Xu et al., 2019; Hamilton et al., 2017; Veličković et al., 2018; Gilmer et al., 2017), along with the state-of-the-art PNA (Corso et al., 2020) and GATv2 (Brody et al., 2022) architectures.

In order to provide a fair comparison we standardize all parameter counts, architectures and optimizers in our experiments. All experiments were run using Adam (Kingma & Ba, 2014). Further details on how we ensured a fair evaluation can be found in the appendix.

We do not use edge features in our experiments as for most baseline architectures there exist no standard method to incorporate them. We do not use sampling, which, as explained in Section 2.2, is ineffective for 4 datasets; for the remaining dataset, Arxiv, we believe it is not in the scientific interest to introduce an additional variable. This also applies to GraphSAGE, where we do not use the commonly applied neighborhood sampling. All experiments were run 10 times.

## 5.2 MAIN RESULTS

Our results across the 5 tasks are shown in Table 1. We draw attention to the following observations:

- **EGC-S is competitive with anisotropic approaches**. We outperform GAT(v1) and MPNN-Sum on all benchmarks, despite our resource efficiency. The clearest exception is MPNN-Max on CIFAR & Code, where the max aggregator provides a stronger inductive bias. We observe that GATv2 improves upon GAT, but does not clearly outperform EGC.

- **EGC-M outperforms PNA**. The addition of multiple aggregator functions improves performance of EGC to beyond that obtained by PNA. We hypothesize that our improved performance over PNA is related to PNA's reliance on multiple degree-scaling transforms. While this approach can boost the representational power of the architecture, we believe that it can result in a tendency to overfit to the training set.

- **EGC performs strongly without running out of memory.** We observe that EGC is one of only three architectures that did not exhaust the VRAM of the popular Nvidia 1080/2080Ti GPUs, with 11GB VRAM, when applied to Arxiv: we had to use an RTX 8000 GPU with 48GB VRAM to run these experiments. PNA, our closest competing technique accuracy-wise, exhausted memory on the Code benchmark as well. Detailed memory consumption figures are provided in Table 4.

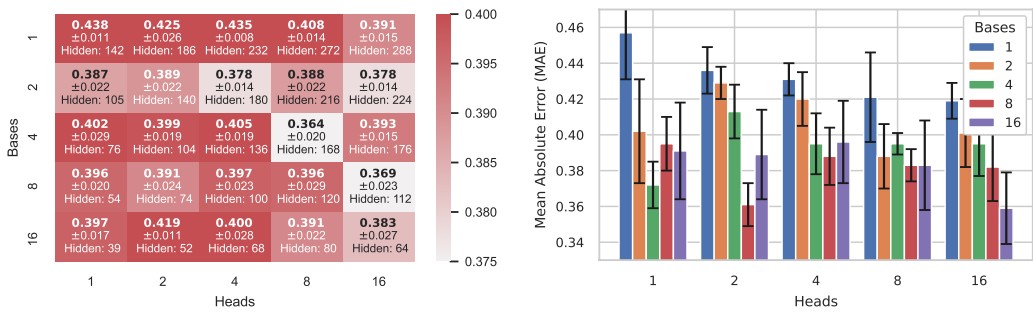

(a) Constant parameter count (100k)  (b) Constant hidden dimension (128)

Figure 3: Study over the number of heads ($H$) and bases ($B$). Study run on ZINC dataset with EGC-S. Metric is MAE (mean $\pm$ standard deviation): lower is better. We study keeping the total parameter count constant, and fixing the hidden dimension. Each experiment was tuned individually. Setting $B > H$ does not necessarily improve performance due to the risk of overfitting, and forces the usage of a smaller hidden dimension to retain a constant parameter count.

Overall, EGC obtains the best performance on 4 out of the 5 main datasets; on the remaining dataset (MolHIV), EGC is the second best architecture. This represents a significant achievement: our architecture demonstrates that we do not need to choose between efficiency and accuracy.

## 5.3 ADDITIONAL STUDIES

**How Should Heads and Bases be Chosen?**    To understand the trade-off between the number of heads ($H$) and bases ($B$), we ran an ablation study on ZINC using EGC-S; this in shown in Figure 3.

The relationship between these parameters is non-trivial. There are several aspects to consider: (1) increasing $H$ and $B$ means that we spend more of our parameter budget to create the combinations, which reduces hidden dimension—as shown in Figure 3. This is exacerbated if we use multiple aggregators: our combination dimension must be $HB|\mathcal{A}|$. (2) Increasing $B$ means we must reduce the hidden size substantially, since it corresponds to adding more weights of size $\frac{F'}{H} \times F$. (3) Increasing $H$ allows us to increase hidden size, since each basis weight becomes smaller. We see in Figure 3 that increasing $B$ beyond $H$ does not yield significant performance improvements: we conjecture that bases begin specializing for individual heads; by sharing, there is a regularizing effect, like observed in Schlichtkrull et al. (2018). This regularization stabilizes the optimization and we observe lower trial variance for smaller $B$.

We advise $B = H$ or $B = \frac{H}{2}$. We find $H = 8$ to be effective with EGC-S; for EGC-M, where more parameters are spent on combination weights, we advise setting $H = 4$. This convention is applied consistently for Table 1; full details are supplied in the appendix and code.

**Should The Combination Weightings ($w$) Be Activated?**    Any activation function will shrink the space the per-node weights $\Theta_h^{(i)}$ can lie in, hence we would expect it to harm performance; this is verified in Table 2. Activating $w$ may improve training stability, but we did not observe to be an issue in our experiments. Another issue is that different aggregators result in outputs with different means and variances (Tailor et al., 2021), hence they need to be scaled by different factors to be combined.

| Activation | EGC-S | EGC-M |
|---|---|---|
| Identity | $0.364 \pm 0.020$ | $0.281 \pm 0.007$ |
| Hardtanh | $0.435 \pm 0.010$ | $0.293 \pm 0.013$ |
| Sigmoid | $0.366 \pm 0.008$ | $0.303 \pm 0.016$ |
| Softmax | $0.404 \pm 0.010$ | $0.307 \pm 0.013$ |

Table 2: Activating the combination weightings $w$ harms performance. Run on ZINC; lower is better.

**Applying EGC to Large-Scale Heterogeneous Graphs**    We evaluated EGC on the heterogeneous OGB-MAG dataset, containing 2M nodes and 21M edges. On a homogeneous version of the graph, we exceed the baselines' performance by 1.5-2%; we experiment with both symmetric normalization (EGC-S) and the mean aggregators to demonstrate that the mechanism utilized by EGC

is effective, regardless of which aggregator provides the stronger inductive bias for the dataset. Our architecture can be expanded to handle different edge types, yielding the R-EGC architecture which improves performance over R-GCN by 0.9%. We expect that accuracy can be further improved by using sampling techniques to regularize the optimization, or using pretrained embeddings; however, adding these techniques makes comparing the results more difficult as it is known that sampling techniques can affect each architecture's performance differently (Liu et al., 2020).

## 5.4 MEMORY AND LATENCY BENCHMARKS

We now assess our model's resource efficiency. For CPU measurements we used an Intel Xeon Gold 5218 and for GPU we used an Nvidia RTX 8000.

**Aggregator Fusion** We evaluated aggregator fusion across several topologies on both CPU and GPU. For space reasons, we leave full details to Appendix D. In summary, we observe that using 3 aggregators over standard SpMM incurs an additional overhead of only **14%**, enabling us to improve model performance without excessive computational overheads at inference time.

**End-to-End Latency** We provide latency and memory statistics for *parameter-normalized* models in Table 4. We draw attention to how slow and memory-intensive the $\mathcal{O}(E)$ models are for training and inference. The reader should note that the inference latency and memory consumption for MPNN and PNA rises by 6-7× relative to EGC, corresponding to the large $\frac{|E|}{|V|}$ ratio for Arxiv. EGC-M offers substantially lower latency and memory consumption than its nearest competitor, PNA, however the precise benefit will be dataset-dependent. Extended results can be found in Appendix E, including results for *accuracy-normalized* models on Arxiv, which further demonstrate EGC's resource efficiency.

| Method | Test Accuracy % ↑ |
|---|---|
| MLP | $26.92 \pm 0.26$ |
| GCN | $30.43 \pm 0.25$ |
| GraphSAGE-Mean | $31.53 \pm 0.15$ |
| EGC-S | $32.13 \pm 0.73$ |
| EGC ($\oplus$ = Mean) | $33.22 \pm 0.50$ |
| R-GCN (Full Batch) | $46.29 \pm 0.45$ |
| R-EGC (Full Batch) | $47.21 \pm 0.40$ |

Table 3: EGC can be applied applied to large scale heterogeneous graphs, outperforming R-GCN by 0.9%.

| Model | Peak Training Memory (MB) | GPU Training Latency (ms) | GPU Inference Latency (ms) |
|---|---|---|---|
| GCN | 1905 | $159.8 \pm 4.6$ | $35.2 \pm 0.1$ |
| GIN | 1756 | $155.2 \pm 3.9$ | $35.2 \pm 0.1$ |
| GraphSAGE | 1352 | $113.9 \pm 5.4$ | $25.1 \pm 0.4$ |
| GAT | 10841 | $324.3 \pm 1.2$ | $84.7 \pm 0.3$ |
| GATv2 | 14124 | $341.8 \pm 0.5$ | $129.2 \pm 0.2$ |
| MPNN-Sum | 14323 | $768.2 \pm 0.8$ | $230.7 \pm 0.3$ |
| MPNN-Max | 14623 | $797.8 \pm 0.9$ | $258.2 \pm 0.4$ |
| PNA | 14533 | $892.7 \pm 1.1$ | $305.7 \pm 0.5$ |
| EGC-S | 2430 | $177.7 \pm 2.2$ | $37.3 \pm 0.1$ |
| EGC-M | 4068 | $220.9 \pm 0.6$ | $42.2 \pm 0.3$ |

Table 4: Memory and latency statistics for *parameter-normalized* models (used in table 1) on Arxiv. Note that EGC-M memory consumption and latency can be reduced with aggregator fusion at inference time.

## 6 DISCUSSION AND CONCLUSION

**How Surprising Are Our Results?** We observed that it was possible to design an isotropic GNN that is competitive with state-of-the-art anisotropic GNN models on 6 benchmarks. This result contradicts common wisdom in the GNN community. However, our results may be viewed as part of a pattern visible across the ML community: both the NLP and computer vision communities have seen works indicating that anisotropy offered by Transformers may not be necessary (Tay et al., 2021; Liu et al., 2022), consistent with our observations. It is worth asking why we observe our results, given that they contradict properties that have been theoretically proven. We believe that there are a variety of reasons, but the most important is that most real world datasets do not require the theoretical power these more expressive models provide to achieve good results. In particular, many real world datasets are homophilous: therefore simplistic approaches, such as EGC, can achieve high performance. This implies that the community should consider adding more difficult datasets to standard benchmarks, such as those presented in Lim et al. (2021) and Veličković et al. (2021).

Our proposed layer, EGC, can be used as a *drop-in replacement* for existing GNN layers, and achieves better results across 6 benchmark datasets compared to strong baselines, with substantially lower resource consumption. Our work raises important questions for the research community, while offering significant practical benefits with regard to resource consumption. We believe the next step for our work is incorporation of edge features e.g. through line graphs (Chen et al., 2020b) or topological message passing (Bodnar et al., 2021).

ACKNOWLEDGEMENTS

This work was supported by the UK's Engineering and Physical Sciences Research Council (EP-SRC) with grant EP/S001530/1 (the MOA project) and the European Research Council (ERC) via the REDIAL project (Grant Agreement ID: 805194). FLO acknowledges funding from the Huawei Studentship at the Department of Computer Science and Technology of the University of Cambridge.

The authors would like to thank Ben Day, Javier Fernandez-Marques, Chaitanya Joshi, Titouan Parcollet, Petar Veličković, and the anonymous reviewers who have provided comments on earlier versions of this work.

ETHICS STATEMENT

The method described in this paper is generic enough that it can be applied to any problem GNNs are applied to. The ethical concerns associated with this work are related to enabling more efficient training and deployment of GNNs. This may be positive (drug discovery) or negative (surveillance), but these concerns are inherent to any work investigating efficiency for machine learning systems.

REPRODUCIBILITY STATEMENT

We have supplied the code required to regenerate our results, along with the hyperparameters required. In addition, we supply pre-trained models. Resources associated with this paper can be found at `https://github.com/shyam196/egc`.

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

| Method | Propagation Rule | Memory | Notes |
|---|---|---|---|
| **GCN** (Kipf & Welling, 2017) | $\mathbf{y}^{(i)} = \mathbf{\Theta} \sum_{j \in \mathcal{N}(i) \cup \{i\}} \frac{1}{\sqrt{\deg(i)\deg(j)}} \mathbf{x}^{(j)}$ | $\mathcal{O}(V)$ | Formally defined for undirected graphs with self-loops; motivated by graph signal processing. |
| **GIN** (Xu et al., 2019) | $\mathbf{y}^{(i)} = f_{\mathbf{\Theta}}[(1+\epsilon)\mathbf{x}^{(i)} + \sum_{j \in \mathcal{N}(i)} \mathbf{x}^{(j)}]$ | $\mathcal{O}(V)$ | $f$ is a learnable function, typically parameterized as an MLP or linear layer; $\epsilon$ may be fixed or learned. |
| **GraphSAGE** (Hamilton et al., 2017) | $\mathbf{y}^{(i)} = \mathbf{\Theta}_1 \mathbf{x}^{(i)} + \bigoplus_{j \in \mathcal{N}(i)} \mathbf{\Theta}_2 \mathbf{x}^{(j)}$ | $\mathcal{O}(V)$ | $\bigoplus$ typically parameterized as mean or max. |
| **GAT** (Veličković et al., 2018) | $\mathbf{y}^{(i)} = \alpha_{i,i} \mathbf{\Theta} \mathbf{x}^{(i)} + \sum_{j \in \mathcal{N}(i)} \alpha_{i,j} \mathbf{\Theta} \mathbf{x}^{(j)}$ | $\mathcal{O}(E)$ | Attention coefficients calculated using: $\alpha_{i,j} = \frac{\exp\left(\text{LeakyReLU}\left(\mathbf{a}^{\top}[\mathbf{\Theta}\mathbf{x}^{(i)} \| \mathbf{\Theta}\mathbf{x}^{(j)}]\right)\right)}{\sum_{k \in \mathcal{N}(i) \cup \{i\}} \exp\left(\text{LeakyReLU}\left(\mathbf{a}^{\top}[\mathbf{\Theta}\mathbf{x}^{(i)} \| \mathbf{\Theta}\mathbf{x}^{(k)}]\right)\right)}$. Common to define multiple attention heads and concatenate. |
| **GATv2** (Brody et al., 2022) | $\mathbf{y}^{(i)} = \alpha_{i,i} \mathbf{\Theta} \mathbf{x}^{(i)} + \sum_{j \in \mathcal{N}(i)} \alpha_{i,j} \mathbf{\Theta} \mathbf{x}^{(j)}$ | $\mathcal{O}(E)$ | Similar to GAT, but with attention calculation redefined to improve expressivity $\alpha_{i,j} = \frac{\exp\left(\mathbf{a}^{\top} \text{LeakyReLU}\left(\mathbf{\Theta}[\mathbf{x}_i \| \mathbf{x}_j]\right)\right)}{\sum_{k \in \mathcal{N}(i) \cup \{i\}} \exp\left(\mathbf{a}^{\top} \text{LeakyReLU}(\mathbf{\Theta}[\mathbf{x}_i \| \mathbf{x}_k])\right)}$. |
| **MPNN** (Gilmer et al., 2017) | $\mathbf{y}^{(i)} = U(\mathbf{x}^{(i)}, \bigoplus_{j \in \mathcal{N}(i)} M(\mathbf{x}^{(i)}, \mathbf{x}^{(j)}, \mathbf{e}_{ij}))$ | $\mathcal{O}(E)$ | $U, M$ typically defined as linear layers acting on concatenated features; $\bigoplus$ may be any valid aggregator, typically sum or max. |
| **PNA** (Corso et al., 2020) | $\mathbf{y}^{(i)} = U(\mathbf{x}^{(i)}, \bigoplus_{j \in \mathcal{N}(i)} M(\mathbf{x}^{(i)}, \mathbf{x}^{(j)}, \mathbf{e}_{ij}))$ | $\mathcal{O}(E)$ | Similar to MPNN, but with $\bigoplus$ defined to use 4 aggregators (mean, standard deviation, max, and min) scaled by 3 different functions of node degree, resulting in 12 different aggregations by default. |

Table 5: Propagation rules for general-purpose GNN architectures we compare against in this work; rules are provided using node-wise formulations. We evaluate against popular architectures, and a recent proposal that has achieved state-of-the-art performance, PNA.

# A    FURTHER EXPERIMENT DETAILS

## A.1    ENSURING FAIRNESS

We expand on our experimental protocol, with a particular focus on describing measures that we took to ensure that the results we report are not unfairly biased towards EGC.

For EGC-S, we use $H = 8$ and $B = 4$, as implied by our ablation for all experiments, with the single exception of OGB-Code, where we use $H = B = 8$. The benefit of using a smaller set of bases is that we can increase the hidden dimension, but this is not viable in this case since most of the 11M parameters in the model correspond to the token read-out layers, which quickly increases as the model hidden dimension grows. As shown by section 5.3, if we cannot increase the hidden dimension, it is better to increase the bases. This is the only exception we make.

For EGC-M, we use $H = B = 4$ for all experiments. The main challenge is aggregator selection, and this remains a major challenge for our work. We were unable to find a satisfactory technique for automated discovery of aggregator choices, hence we rely on heuristics to find them. We restrict ourselves to use 3 aggregators for each model (yielding 35 possible choices). In order to determine the aggregators, we use two heuristics: (1) aggregators should be "diverse" and (2) aggregators should be chosen based on inductive bias for task. Using these two rules, we try up to 3 possible choices of aggregators; all choices considered are shown in Table 6. We note that while some choices do improve performance, our conclusions are not invalidated; it is also worth noting that it is likely that better aggregator choices can be found.

## A.2    CLUSTER DETAILS

Most of our experiments were run on several machines in our SLURM cluster using Intel CPUs and NVIDIA GPUs. Each machine was running Ubuntu 18.04. The GPU models in our cluster were RTX 2080Ti and GTX 1080Ti. High-memory experiments were run on V100s in our cluster and an RTX 8000 virtual machine we had access to.

| Dataset | Add | Mean | Symmetric Normalization | Max | Min | Std | Var | Result |
|---|---|---|---|---|---|---|---|---|
| **Zinc** | ✓ | | | ✓ | | ✓ | | $0.281 \pm 0.007$ |
| **Zinc** | | | ✓ | ✓ | ✓ | | | $0.284 \pm 0.045$ |
| **CIFAR** | ✓ | | | ✓ | | ✓ | | $71.03 \pm 0.42$ |
| **CIFAR** | | | ✓ | ✓ | ✓ | | | $70.05 \pm 1.14$ |
| **MolHIV** | ✓ | ✓ | | ✓ | | | | $78.19 \pm 1.54$ |
| **MolHIV** | ✓ | | | ✓ | | | ✓ | $77.40 \pm 1.02$ |
| **MolHIV** | ✓ | | | ✓ | | ✓ | | $77.98 \pm 1.65$ |
| **Arxiv** | | ✓ | ✓ | ✓ | | | | $71.96 \pm 0.23$ |
| **Arxiv** | ✓ | ✓ | ✓ | | | | | $70.59 \pm 0.67$ |
| **Arxiv** | ✓ | ✓ | | ✓ | | | | $70.38 \pm 0.76$ |
| **Code-V2** | | | ✓ | ✓ | ✓ | | | $0.1595 \pm 0.0019$ |
| **Code-V2** | | | | ✓ | ✓ | ✓ | | $0.1572 \pm 0.0029$ |
| **Code-V2** | ✓ | | | ✓ | | ✓ | | $0.1578 \pm 0.0021$ |

Table 6: Possible aggregators tried for EGC-M. Up to 3 combinations (from a possible 35) were tried, as we limited ourselves to always using 3 aggregators. Our conclusions do not change, and it is likely that better results can be found with more optimal aggregator choices.

| Model | ZINC (MAE $\downarrow$) | CIFAR (Acc. $\uparrow$) | MolHIV (ROC-AUC $\uparrow$) | Code-V2 (F1 $\uparrow$) |
|---|---|---|---|---|
| **GA-MLP** | $0.510 \pm 0.037$ | $58.13 \pm 0.65$ | $75.50 \pm 1.32$ | $0.1485 \pm 0.0027$ |
| **GCN** | $0.459 \pm 0.006$ | $55.71 \pm 0.38$ | $76.14 \pm 1.29$ | $0.1480 \pm 0.0018$ |
| **EGC-S** | $0.364 \pm 0.020$ | $66.92 \pm 0.37$ | $77.44 \pm 1.08$ | $0.1528 \pm 0.0025$ |

Table 7: Results of applying graph-augmented MLPs (GA-MLPs) to tasks requiring generalization to unseen graphs. We see that the performance is broadly similar to the corresponding GCN model—and far weaker than EGC-S, which uses the same aggregator. We also considered using the add aggregation on ZINC, and achieved $0.444 \pm 0.019$. By comparison, the equivalent GIN model (which uses the add aggregation) achieved $0.387 \pm 0.015$.

## B  LIMITATIONS OF EXISTING APPROACHES

As explained in the related work, existing approaches to improving GNN efficiency have severe limitations. In this section we elaborate upon them, and provide experimental evidence where necessary.

We first examine graph-augmented MLPs (GA-MLPs); to our knowledge there are few experimental results assessing the performance of these models when they are applied to problems requiring generalization to unseen graphs. In the literature they are generally applied to large scale node classification benchmarks, such as those found in the OGB benchmark suite (Wu et al., 2019b; Rossi et al., 2020). It is known that these models are theoretically less expressive than standard GNNs, however their performance when applied to node classification datasets has been acceptable.

We consider models of the form:

$$\mathbf{Y} = \text{Readout}(\text{MLP}( \overset{4}{\underset{k=0}{\big\|}} \mathbf{S}^k \mathbf{X} \mathbf{W})) \qquad (5)$$

We use up to the 4th power of the diffusion operator $\mathbf{S}$ to emulate the depth of the corresponding GNNs, which all use 4 layers. We set $\mathbf{S}$ to use symmetric normalization, as used by GCN and EGC-S; we also consider setting $\mathbf{S}$ to the adjacency matrix on ZINC, to emulate the operations used by GIN. The results are provided in Table 7. We see that the GA-MLP models offer similar (but often worse) performance than the corresponding GNN baselines. The models are not competitive with approaches such as GAT or MPNN, and is outperformed by a wide margin by EGC-S. Achieving

| Experiment | ZINC (MAE ↓) | CIFAR (Acc. ↑) | MolHIV (ROC-AUC ↑) | Code-V2 (F1 ↑) |
|---|---|---|---|---|
| EGC-S | $0.364 \pm 0.020$ | $66.92 \pm 0.37$ | $77.44 \pm 1.08$ | $0.1528 \pm 0.0025$ |
| EGC-S + DropEdge ($p = 0.1$) | $0.468 \pm 0.007$ | $66.37 \pm 0.28$ | $77.41 \pm 1.32$ | $0.1553 \pm 0.0021$ |
| EGC-S + DropEdge ($p = 0.5$) | $0.629 \pm 0.023$ | $64.60 \pm 0.58$ | $75.33 \pm 0.82$ | $0.1527 \pm 0.0019$ |
| EGC-S + GraphSAINT Node Sampler ($p = 0.1$) | $0.631 \pm 0.012$ | $61.37 \pm 0.75$ | $73.65 \pm 1.41$ | $0.1461 \pm 0.0027$ |

Table 8: Results of applying sampling approaches to EGC-S. We do not enable sampling at test time—hence these approaches do not offer any test time reductions to computation. Sampling, in principle, is applicable to any underlying GNN: our conclusions will transfer to other underlying GNNs. We see that DropEdge (Rong et al., 2020) with a low drop probability ($p = 0.1$) can aid model performance; however, setting $p$ this low does not significantly reduce memory consumption or computation. Increasing $p$ to 0.5 does reduce resource consumption noticeably, but results in noticeable degradation to model performance.

state-of-the-art performance with GA-MLPs does not appear to be possible, at least with our current understanding of these models.

We now proceed to investigate sampling, in which each training step does not run over the entire graph, but over some sampled subgraph. These methods have seen great popularity when applied to tasks such as node classification, and they are able to deliver sampling ratios in excess of $20\times$, hence yielding noticeable improvements to memory consumption (the primary limitation with large scale training). However, we note that the benefit of these methods has not been examined carefully for many graph topologies; while they have been shown to be effective for many "small-world" graph topologies—which arise in many graphs—it is not the case that all graphs fall into this category. For example, molecule graphs would not fit this category.

We first assess the sampling strategy from GraphSAINT (Zeng et al., 2019) by applying it to EGC-S models. For our experiments, we use the node-centric sampler. The results are presented in Table 8; we disable sampling at test time. Even with a relatively low dropping probability of 10% (i.e. 90% of nodes are retained), the model performance degradation is severe. We note that the computational savings achieved are modest when using this drop probablility. We also observed similar results when using the edge-centric sampler from GraphSAINT.

We also attempted a different approach for sampling proposed by DropEdge (Rong et al., 2020); as before, we apply it to EGC-S models. In this simple scheme, elements of the adjacency matrix are dropped; this scheme was shown to be effective for training deep graph networks, but it is also useful for reducing the computational footprint of models, since it effectively reduces the number of messages that have to be computed. We also provide results in Table 8. The results are significantly better than we observe when using GraphSAINT's sampling strategies: at 10% drop probabilities we even see an improvement in some cases, due to the regularization effect. However, once we increase the drop probability to levels where we would observe a noticeable reduction in computational demand, we observe that model performance declines. In summary, while DropEdge is a more effective strategy for sampling on many inductive tasks, it is not beneficial as a method to reduce the computational burden.

Finally, we discuss the limitations of other approaches proposed in the literature. Quantization is an approach that is typically applied at inference time; mixed-precision approaches can be applied at training time, however care must be taken for GNNs to avoid biasing the gradients (Tailor et al., 2021). Additionally, while neural architecture search (NAS) may be useful to find memory-efficient models if the search objective is set appropriately, they suffer from some limitations—primarily search time and memory consumption. Finally, approaches such as reversible residuals (Li et al., 2021) are useful to architecture design, they do not tackle issues such as high peak memory usage induced by the message passing step.

## C  APPROACHES FOR HARDWARE ACCELERATION OF GNNS

The reader should note that most existing work for GNN hardware acceleration focuses on supporting only a subset of GNNs: specifically, they tend to only support models that can be implemented

using SpMM. Approaches in the literature falling into this area include Chen et al. (2020a), Yan et al. (2020), You et al. (2021) and Zhang et al. (2021). It is possible to add greater flexibility to the accelerator to support more expressive message passing schemes, however this necessarily implies greater complexity. As Amdahl's law (Amdahl, 1967) implies, increasing flexibility is likely to reduce peak performance, while increasing silicon area requirements. Therefore, aiming for the simplest primitive (as we do with EGC) is the most sensible approach to obtain hardware acceleration.

## D  AGGREGATOR FUSION

---

**Algorithm 1** Aggregator Fusion with aggregators $\mathcal{A}$. This method is a modification of the Compressed Sparse Row (CSR) SpMM algorithm, where we maximize re-use of matrix $\mathbf{B}$. Maximizing re-use enables us to obtain significantly better accuracy with minimal impact on memory and latency. For simplicity, pseudocode assumes $H = B = 1$. This version demonstrates how we can remove memory overheads at inference time.

---

**Input:** CSR $\mathbf{A} \in \mathbb{R}^{N \times N}$, Dense $\mathbf{B} \in \mathbb{R}^{N \times F}$, Combination weightings $\mathbf{w} \in \mathbb{R}^{N \times |\mathcal{A}|}$
**Output:** Dense $\mathbf{C} \in \mathbb{R}^{N \times F}$
**for** $i = 0$ **to** $\mathbf{A}$.rows $- 1$ **do**
    **for** $jj = \mathbf{A}$.row_pointer$[i]$ **to** $\mathbf{A}$.row_pointer$[i + 1]$ **do**
        $j = \mathbf{A}$.column_index$[jj]$
        Init temp arrays of length $F$ per aggregator
        $a_{ij} = \mathbf{A}$.values$[jj]$
        {May be faster to interleave these calls:}
        **for** $\oplus \in \mathcal{A}$ **do**
            process_row$_{\oplus}(a_{ij}, \mathbf{B}[i, :], \text{temp}_{\oplus})$
        **end for**
    **end for**
    {Can be generalized to $H, B > 1$:}
    $\mathbf{C}[i, :] = \sum_{\oplus \in \mathcal{A}} \mathbf{w}[i, \oplus] \cdot \text{temp}_{\oplus}[:]$
**end for**

---

The naive approach of performing each aggregation sequentially would cause a linear increase in latency with respect to $|\mathcal{A}|$. However, a key observation to note is that we are *memory-bound*: the bottleneck with sparse operations is waiting for the data to arrive from memory. This observation applies to both GPUs and CPUs, and justified through profiling. Using a profiler on a GTX 1080Ti we observed that SpMM using the Reddit graph Hamilton et al. (2017) with feature sizes of 256 achieved just 1.2% of the GPU's peak FLOPS, with 88.5% of stalls being caused by unmet memory dependencies. The fastest processing order performs as much work as possible with data that has already been fetched from memory, rather than fetching it multiple times. This concept is illustrated in Algorithm 1 in the appendix. We can perform all aggregations as a lightweight modification to the standard compressed sparse row (CSR) SpMM algorithm.

The second observation we make is that storing the results of all aggregations is unnecessary at inference time. Note that the CSR SpMM algorithm processes each row in the output matrix sequentially: rather than storing the aggregations for every row, instead we should store only the weighted results. This approach not only reduces memory consumption, but also latency as we improve the effectiveness of our cache and reduce memory system contention. In practice, this optimization is especially important when performing inference on topologies which have more frequent periods where the processing has become compute-bound, since we reduce contention between load and store units Corporation. This is also demonstrated in Algorithm 1.

We evaluated aggregator fusion across four different topologies, on both CPU and GPU; our results can be found in Table 9. We assumed all operations are 32-bit floating point, and that we were using three aggregators: summation-based, max, and min; these aggregators match those used for EGC-M Code. Our benchmarks were conducted on a batch of 10k graphs from the ZINC and Code datasets, Arxiv, and the popular Reddit dataset (Hamilton et al., 2017), which is one of the largest graph datasets commonly evaluated on in the GNN literature. Our SpMM implementation on GPU is based on Yang et al. (2018). Code for the kernels are provided in our repo.

| Method | CPU (Xeon Gold 5218) | | | | GPU (RTX 8000) | | | |
|---|---|---|---|---|---|---|---|---|
| | Reddit / s | Code / s | Arxiv / s | ZINC / s | Reddit / ms | Code / ms | Arxiv / ms | ZINC / ms |
| Weight Matmul | $0.07 \pm 0.00$ | $0.423 \pm 0.023$ | $0.055 \pm 0.013$ | $0.074 \pm 0.010$ | $2.36 \pm 0.00$ | $13.66 \pm 0.12$ | $1.74 \pm 0.01$ | $2.36 \pm 0.02$ |
| CSR SpMM | $29.08 \pm 0.12$ | $1.943 \pm 0.040$ | $0.760 \pm 0.021$ | $0.315 \pm 0.006$ | $186.44 \pm 0.05$ | $19.88 \pm 0.10$ | $5.56 \pm 0.02$ | $3.39 \pm 0.01$ |
| Naive Fusion | $88.25 \pm 0.20$ | $8.680 \pm 0.094$ | $2.631 \pm 0.016$ | $1.482 \pm 0.010$ | $595.91 \pm 0.13$ | $112.52 \pm 0.22$ | $23.81 \pm 0.06$ | $19.09 \pm 0.05$ |
| + Faster Ordering | $40.05 \pm 0.10$ | $4.592 \pm 0.054$ | $1.303 \pm 0.037$ | $0.772 \pm 0.008$ | $214.60 \pm 0.10$ | $29.38 \pm 0.12$ | $7.63 \pm 0.03$ | $4.78 \pm 0.02$ |
| + Store Weighted Result Only | $38.63 \pm 0.22$ | $1.752 \pm 0.018$ | $0.952 \pm 0.026$ | $0.278 \pm 0.003$ | $208.22 \pm 0.13$ | $26.64 \pm 0.09$ | $6.75 \pm 0.03$ | $4.38 \pm 0.02$ |

Table 9: Inference latency (mean and standard deviation) for CSR SpMM, used by GCN/GIN, and aggregator fusion. Assuming a feature dimension of 256 and $H = B = 1$ per Algorithm 1. We observe that aggregator fusion results in an increase of 34% in the worse case; in contrast, the naive implementation has a worst case increase of 466%. Also included are timings for dense multiplication with a square weight matrix; we observe that sparse operations dominate latency measurements.

| Model | Train Epoch Time (s) | Test Epoch Time (s) | Peak Train Memory (MB) |
|---|---|---|---|
| GCN | $144.7 \pm 0.5$ | $6.3 \pm 0.3$ | 1337 |
| GIN | $134.5 \pm 0.1$ | $5.7 \pm 0.3$ | 1331 |
| GraphSAGE | $140.3 \pm 0.3$ | $5.6 \pm 0.3$ | 1226 |
| GAT | $176.0 \pm 0.7$ | $6.3 \pm 0.3$ | 2885 |
| MPNN-Sum | $305.3 \pm 2.7$ | $6.8 \pm 0.3$ | 3448 |
| MPNN-Max | $319.0 \pm 1.3$ | $7.6 \pm 0.4$ | 3901 |
| PNA | $575.4 \pm 2.3$ | $12.0 \pm 0.2$ | 9399 |
| EGC-S | $166.4 \pm 2.7$ | $6.2 \pm 0.3$ | 1842 |
| EGC-M | $225.7 \pm 0.6$ | $7.9 \pm 0.3$ | 3470 |

Table 10: Latency and memory results for *parameter-normalized* models on OGB Code-V2. Despite having a lower $\frac{|E|}{|V|}$ ratio of 2.75 relative to Arxiv (13.67), we see that the trends we observed in for Arxiv broadly remain. It is worth noting again that EGC-M is far more efficient both latency and memory-wise than PNA.

As expected, our technique optimizing for input re-use achieves significantly lower inference latency than the naive approach to applying multiple aggregators. While the naive approach results in a mean increase in latency of 331%, our approach incurs a mean increase of **only 14%** relative to ordinary SpMM, used by GCN and GIN. The increase is topology dependent, with larger increases in latency being observed for topologies which are less memory-bound. We also provide timings for dense matrix multiplication (i.e. $\mathbf{X\Theta}$) to justify our focus on optimizing sparse operations in this work: CSR SpMM operation is **4.7×** slower (geomean) than the corresponding weight multiplication. We believe further optimizations of the operations used by architecture are achievable through the use of auto-tuning frameworks e.g. TVM (Chen et al., 2018b), but this lies beyond the scope of this work.

## E    LATENCY AND MEMORY CONSUMPTION ON OTHER DATASETS

In the evaluation, we assessed the memory consumption and latency for the parameter-normalized models on Arxiv. In this section, we consider a similar exercise for OGB Code models. The results are provided in Table 10. Our conclusions remain broadly similar, with EGC-M offering clear improvements to memory consumption, latency, and parameter efficiency relative to PNA. EGC-S is superior to GAT, with similar inference latency, better model performance, and noticeably lower memory consumption. We train models using batch size 128; the reader should note that the memory consumption figures can vary between runs (even for the same model), since graphs vary in the number of nodes and edges.

So far we have not demonstrated that our approach is more efficient than the baselines on Arxiv, which is the only dataset where EGC-M and PNA are not the best performing. To demonstrate that we are more efficient we must show that we achieve lower memory consumption and latency for a given accuracy-level—i.e. we must consider models that are *accuracy-normalized*. We evaluate increasing the parameter count for baseline models until they achieve the same accuracy as EGC-S;

| Accuracy-Normalized Model | Parameters | GPU Training Latency (ms) | GPU Inference Latency (ms) | Peak Memory (MB) |
|---|---|---|---|---|
| GCN | 184k | 208.6 ± 2.2 | 44.9 ± 0.4 | 2549 |
| GIN | *FAIL* | *FAIL* | *FAIL* | *FAIL* |
| GraphSAGE | 593k | 335.8 ± 2.4 | 60.2 ± 0.2 | 3208 |
| EGC-S | 100k | 177.7 ± 2.2 | 37.3 ± 0.1 | 2430 |

Table 11: Latency and memory statistics for ***accuracy-normalized*** models on Arxiv. To achieve the same accuracy as EGC-S, we must boost the size of the baseline models. We observe that EGC-S offers noticeable reductions to both memory consumption and latency for a given accuracy level.

we also gave these models an extra advantage over our method by increasing the hyperparameter search budget. The results are shown in Table 11, where we observe that EGC-S is more efficient once we are comparing models achieving the same accuracy.

The reader should note that GAT (but not GATv2) can be implemented to reduce memory consumption by noting that the left and right halves of the attention vector can be computed separately, and added together as appropriate. We use an optimized implementation of GAT for our experiments (from PyTorch Geometric); we refer the reader to the implementation for further details.

## F   GENERALIZING TO HETEROGENEOUS GRAPHS

Our R-EGC model is similar to the baseline R-GCN model included in the OGB repository. The OGB model deviates from the standard definition of R-GCN (Schlichtkrull et al., 2018) since it handles different *node* types, not just edge types. The baseline model has weights to generate messages for each relation-type, and a weight matrix to update each individual node type. This corresponds to:

$$\mathbf{y}^{(i)} = \boldsymbol{\Theta}_\eta \mathbf{x}^{(i)} + \sum_{r \in \mathcal{R}} \frac{1}{|\mathcal{N}_r(i)|} \sum_{j \in \mathcal{N}_r(i)} \boldsymbol{\Theta}_r \mathbf{x}^{(j)} \tag{6}$$

where $\eta$ corresponds to the *node type* of node $i$, and $\mathcal{R}$ represents the set of relation types. Note that the mean aggregator is used.

We deviate from the baseline by using a single set of basis weights. Instead, we use a different weighting calculation layers ($\mathbf{w}^{(i)} = \boldsymbol{\Phi}\mathbf{x} + \mathbf{b}$) per node and relation type.

$$\mathbf{y}^{(i)} = \bigg\|_{h=1}^{H} \sum_{b=1}^{B} w_{\eta,h,b}^{(i)} \boldsymbol{\Theta}_b \mathbf{x}^{(j)} + \sum_{r \in \mathcal{R}} \frac{1}{|\mathcal{N}_r(i)|} \bigg\|_{h=1}^{H} \sum_{b=1}^{B} w_{r,h,b}^{(i)} \sum_{j \in \mathcal{N}(i)} \boldsymbol{\Theta}_b \mathbf{x}^{(j)} \tag{7}$$

