# OpenReview forum: "Do We Need Anisotropic Graph Neural Networks?"
_ICLR.cc/2022/Conference — ICLR 2022 Poster_

### Official Review · Reviewer_x8K1 · 2021-10-31

**Correctness:** 3
**Technical Novelty And Significance:** 3
**Empirical Novelty And Significance:** 4
**Recommendation:** 8
**Confidence:** 3

**Main Review:**

I am convinced by the introduction part. The hardware efficiency and memory cost are arguably two bottlenecks in the GCN inference and training and they are often correlated.

Strengths:

* The recap of algorithm-hardware co-design is clean and to the point and should promote the understanding of hardware acceleration for the GCN community, potential reference also includes GCoD (HPCA'22) and G-CoS (ICCAD'21).

* The comparison table give the readers high-level information about the propagation differences among various GCN methods and their corresponding memory requirements.

* The adaptive filter is a kind of meta-learning approach that allows different basis filters to explore different latent spaces and finally weighted summed. I suppose the experiments are a fair comparison with a similar number of parameters as Fig. 3 shows.

* A clean codebase is provided.

Questions:

* I have one question about the sampling-based methods. As Table 2 shows, GraphsSAGE achieves the lowest GPU training and inference latency and also the least peak training memory, which seems to contradict the argument in Sec. 2 that sampling-based methods are often ineffective. In addition, why the inference time of GraphSAGE is also lower than GCN since we will not sample the subgraphs during inference? I am also wondering whether EGC can also be implemented in a sampling-based way?

* Could you elaborate more about the potential impact on the GCN hardware acceleration? Will EGC-S or EGC-M propose unique challenges/opportunities that commercial devices (CPU or GPU) cannot efficiently handle while needing further customized hardware architecture for leveraging their full potential?

**Summary Of The Paper:**

This paper investigates whether incorporating anisotropy (treating neighbors differently using latent functions and thus resulting in $\mathcal{O}(E)$ memory cost) is necessary for boosting GCN's accuracy. They argue that the proposed EGC with $\mathcal{O}(V)$ memory requirement can achieve higher accuracy than prior anisotropy-based works (e.g., GAT) using adaptive filters and thus achieve (1) better accuracy than vanilla GCN; (2) and less memory cost and latency than anisotropy-based methods.

**Summary Of The Review:**

I think the introduction and related works description clearly recap the background of both GCN algorithm and potential algorithm-accelerator co-design, and also the current dilemma in the GCN community, that is, the more accurate but less time/memory efficient anisotropy approaches VS. less accurate but more time/memory efficient vanilla GCN approaches. The proposed EGC can alleviate such a dilemma by achieving both higher accuracy and efficiency.

---

> ### Author Response · Authors · 2021-11-18
> **Response to x8K1**
>
> Thank you for your kind review! We hope we have addressed your questions below :-)
>
> > reference also includes GCoD (HPCA'22) and G-CoS (ICCAD'21).
>
> Thank you. We have added both. An earlier draft of this work included a longer discussion in the background regarding hardware acceleration; we have added this back, but for space reasons we have had to relegate it to the appendix.
>
> > A clean codebase is provided.
>
> Thanks for noting this! We’re also proud to have included pre-trained models.
>
> > In addition, why the inference time of GraphSAGE is also lower than GCN since we will not sample the subgraphs during inference
>
> To be clear, we are examining the GraphSAGE message function, and not considering sampling when implementing the architecture. As a result, there is no overhead to sampling in any of our experiments. This is admittedly unclear, and we have added a comment to the manuscript addressing this.
>
> >  As Table 2 shows, GraphsSAGE achieves the lowest GPU training and inference latency and also the least peak training memory
>
> This is not due to sampling, but instead due to the way the GraphSAGE message function is defined. We have to define two weight matrices W_1 and W_2 of the same size. This means for an equivalent parameter count (relative to GCN, GIN or EGC-S) we must have a smaller hidden size for GraphSAGE. This therefore results in smaller memory consumption and inference latency at the same parameter count. If we were to use the same hidden size, we would not see this effect.
>
> You will find our response to reviewer Cq1y to have memory statistics when we fix the hidden dimension. The observed difference disappears.
>
> > I am also wondering whether EGC can also be implemented in a sampling-based way?
>
> Yes -- the design of the message function (which is what our work focuses on) is orthogonal to the choice of how the data is loaded. You can use a method e.g. GraphSAINT to prepare mini-batches for an EGC-based architecture. This is a common question and we have added a comment to our paper addressing this specific question.
>
> > Will EGC-S or EGC-M propose unique challenges/opportunities that commercial devices (CPU or GPU) cannot efficiently handle while needing further customized hardware architecture for leveraging their full potential?
>
> Thanks for the question. Our architecture relies on SpMM, which is a kernel that has been well studied for decades, and for which we have good kernels for both CPU and GPU. However, we would benefit from dedicated SpMM acceleration in hardware. Aggregator fusion is a relatively simple idea that should not require sophisticated hardware to implement. We are beginning to see sparsity support implemented into real world accelerators, albeit primarily for supporting unstructured sparsity in other types of architectures.
>
> An important observation is that hardware acceleration for very expressive GNNs is rare in the literature. Most works focus on supporting SpMM-based models, which EGC falls into, and can therefore be run on existing proposed accelerators.

---

### Official Review · Reviewer_Cq1y · 2021-11-03

**Correctness:** 3
**Technical Novelty And Significance:** 3
**Empirical Novelty And Significance:** 2
**Recommendation:** 6
**Confidence:** 3

**Main Review:**

This paper is a mixed bag. On one hand, I appreciate the authors' vision: take some things we've learned about GNNs (heads, aggregators), revisit first principles (isotropic GNNs), try to come up with something new that blends them. In addition, I appreciate their take on computational efficiency: they correctly seize upon the notion that a well-understood algorithmic kernel can have meaningful computational impact, and they build on its advantages (e.g., section 3.2-aggregator fusion). On the other hand, many of the conclusions leave something to be desired. The analyses and descriptions sometimes gloss over important factors, and I wish the results delivered a stronger takeaway to match the ambition. For all the talk about computation and memory efficiency, the actual numbers are not all that substantial.

Strengths:

- The authors' direct challenge to accepted folk knowledge about anisotropic GNNs is healthy and welcome. As models evolve over time, it is important for the community to revisit fundamental positions on model components. The paper opens a lot of areas for other researchers to build off it, which I see as a major strength.
- The proposed model is clearly explained and fairly easy to follow. The intrepretations are useful and provide good intuition for how to place their proposed model in the context of other current work.
- The appendices actually provide a lot of useful content, sometimes more than the paper itself. The rationale in appendix C, for instance, was far clearer than the paper's 'aggregator fusion' section in 3.2 (even without the algorithm).

Weaknesses:

- Some of the evaluation results are weak. The accuracy-normalized results in seem to demonstrate that EGC-S  is not all that much faster or smaller than GCN. It also seems to indicate that parameter-normalized results do not correlate particularly well with memory footprint across models, which weakens the argument for O(V) vs. O(E).
- Despite a nominally primary argument of the paper being about memory efficiency and computational performance, very little quantitative results support that argument. I agree that there is definitely a lot of potential for this approach to be efficient, but I wish the paper had strong support for it.

- The authors fail to disambiguate different types of memory effects. For instance, while high-watermark memory footprint (which is mostly what is measured in their experiments) is important, they overlook the effects of parallelism. This leads them, amongst other things, to dismiss sampling methods almost out of hand. The memory behavior of sampled methods is vastly different than non-sampled GNNs when dealing with distributed execution or on graphs that are substantially larger than single-device memory capacity.  Or the focus on O(V)/O(E), which glosses over the fact the average degree is small (between 2 and 13 on their chosen OGB datasets) and ignores other 'constants' on the same order. This is technically correct but misleading. None of these invalidate the authors' methods, but it's somewhat disingenuous.

Also, it very much felt like parts of the paper were aimed at different goals. The intro and background lean heavily into hardware efficiency, while other parts of the paper (S3 outside of the last paragraph; S5.1-5.3) seem to ignore it.

**Summary Of The Paper:**

The authors propose a different formulation for a graph neural network, focused on achieving accuracy equal to anisotropic GNNs without anisotropic mechanisms.

**Summary Of The Review:**

The authors take a different tact with their proposed GNN, which is welcome. They explain it clearly, and the paper's writing and organization are solid. The results support their claim that they can achieve competitive accuracy, but the quantitative computation and memory results are somewhat underwhelming. In my view, the contribution of this paper is its approach and challenge to convention, which feels good enough to publish, even if it left me wanting more substantive.

---

> ### Author Response · Authors · 2021-11-18
> **Response to Cq1y**
>
> Thank you for your kind and thoughtful review. We will try to address your concerns below.
>
> > The authors' direct challenge to accepted folk knowledge about anisotropic GNNs is healthy and welcome
>
> We appreciate that you’ve made this observation. We think the topic of this paper is very important for the community to improve models further.
>
> > The accuracy-normalized results in seem to demonstrate that EGC-S is not all that much faster or smaller than GCN
>
> We have updated the manuscript to emphasize that the accuracy normalized results have been run on Arxiv, where the baseline O(V) models are most competitive (see the paragraph before the conclusion). In other cases (other than MolHIV) we are clearly better than the competition -- as the best model is typically EGC-M, which outperforms the inefficient PNA model.
>
> To be clear: the Arxiv results are a “worst case” -- and even in that situation, we see ~20% reductions in latency, which is still worthwhile. We feel uncomfortable claiming that we are more efficient without demonstrating that we achieve the same accuracy for less cost (or equivalently, more accuracy for the same cost).
>
> > parameter-normalized results do not correlate particularly well with memory footprint across models, which weakens the argument for O(V) vs. O(E).
>
> This is due to the variance in the hidden dimension across different models. If we fix the hidden dimension to 128, we get the following results on Arxiv:
>
> | Model    | Peak Memory / MB |
> |----------|------------------|
> | GCN      | 1624             |
> | GIN      | 1476             |
> | SAGE     | 1480             |
> | GAT      | 9300             |
> | MPNN-MAX | 16120            |
> | MPNN-SUM | 15790            |
> | PNA      | 24372            |
> | EGC-S    | 1815             |
> | EGC-M    | 3803             |
> We can see that O(V) models use roughly similar memory, and significantly less than the O(E) models. There is a small overhead to EGC-S compared to GCN, but as we argue in the paper, this does not appear when we increase the size of the GCN model to achieve the same accuracy.
>
> > Despite a nominally primary argument of the paper being about memory efficiency and computational performance, very little quantitative results support that argument.
>
> We believe that the main practical takeaway from this paper for most users will be the effectiveness of EGC-M. In this context, we would argue that the major efficiency gains over PNA are certainly valuable. In support of this please see the results in Table 10 (in the appendix).
>
> We see that the memory consumption for EGC-M is around a third of PNA’s, along with the training epoch time being less than half before. These improvements are achieved while we also improve model performance.
>
> Please note that peak memory consumption can vary depending on the mini-batches (hence you can get unlucky mini-batches causing OOMs for PNA). This is why we focus on using full-batch Arxiv for our results in the main paper.
>
> > This leads them, amongst other things, to dismiss sampling methods almost out of hand. The memory behavior of sampled methods is vastly different than non-sampled GNNs when dealing with distributed execution or on graphs that are substantially larger than single-device memory capacity
>
> Thank you for mentioning this. It is not our intention to imply that our method is a replacement for sampling methods. It is merely an orthogonal approach investigating efficient architecture design from first principles. Both our method and sampling can be usefully combined. We focus on the benefits of efficient architecture design in particular in this paper -- such as when we need to deal with unseen graphs.
>
> We have updated the manuscript to make this clearer to readers.
>
> > ignores other 'constants' on the same order. This is technically correct but misleading
>
> We have updated the manuscript to be clearer that our method’s benefit is a function of the dataset properties.
>
> However, we would argue that the typical memory reduction of $\geq$2x is still very noteworthy and useful in practice. The benefits as the average node degree increases can be substantial. See [1] where architectural simplification was applied to point cloud GNNs, yielding latency reductions of up to 4.5x (along with the corresponding memory reductions).
>
> > Also, it very much felt like parts of the paper were aimed at different goals. The intro and background lean heavily into hardware efficiency, while other parts of the paper (S3 outside of the last paragraph; S5.1-5.3) seem to ignore it.
>
> We will update the writing for the camera ready to improve the coherence of the paper. Thank you for the specific suggestions!
>
> [1] “Towards Efficient Point Cloud Graph Neural Networks Through Architectural Simplification” by Tailor et al.

---

### Official Review · Reviewer_gVtv · 2021-11-08

**Correctness:** 3
**Technical Novelty And Significance:** 3
**Empirical Novelty And Significance:** 3
**Recommendation:** 6
**Confidence:** 3

**Main Review:**

Strengths:
- The paper conducts a thorough analysis and comparison of alternative approaches and describes how some of the motivations behind the alternatives can be gained without the memory costs.
- The paper conducts a thorough analysis on latency, which I appreciate.
- I like that the authors consider hardware - in many cases, more efficient algorithms in FLOP terms are less efficient because they are not parallelisable. This is an important point and I encourage work in the GNN literature that addresses this.
- The results are better than the baselines, which are over a number of different graph networks. I think the model description is quite general.

Weaknesses:
- To state the obvious, such a network is only really applicable to data where you do not need to consider edge features explicitly. While this is fine in principle, the paper doesn't really make this distinction and so the claim is probably broader than the evidence supports.
- The text seems to imply that the computational efficiencies derive from the use of sparse matrix multiplies. But, many new accelerators do not support sparse matmuls well at all, instead are very strong on dense compute. This distinction should be clearer in the text - this is a *GPU* optimised GNN, not an accelerator optimised GNN (perhaps a more parallel GNN would count for something more general purpose).
- I do not think these results are state of the art - simply that they are better than the baselines reported. Please remove the phrase state of the art.
- Whilst I understand that this method is different from the literature, it is *very* related to concepts such as attention. Thus, I think the impact may be limited.

Some nits: It would be nice to have more detail on the OGB and Zinc results - perhaps explain why this model does not perform as well as MolHIV.





**Summary Of The Paper:**

This paper introduces Adaptive Filters that enable some of the benefits of Message Passing architectures, but while maintaining a memory consumption that scales with the number of nodes.

The authors claim that this architecture is not only better performing, and use less memory, but more efficient for GPUs through the use of sparse matrix multiplication.

The idea of the model is that you have a number of filters (MLPs or linear layers) applied to each sending node latent, a "filter". These are then weighted and summed by a weighting vector calculated as a function of the receiving node. Since there are no functions that take as input both sending, receiving or edge inputs, the memory will scale with the number of nodes. You, in essence, get an efficient pseudo-attention mechanism.


**Summary Of The Review:**

I think this is a thorough piece of work, but I am concerned that the impact is limited for the following reasons:
- I am not convinced that the results presented reflect the current state of of O(V) scaling GNNs. You cite "Training graph neural networks with 1000 layers" which I believe scales O(V), and I think achieves better results on OGBN-ARXIV.
- The ideas seem very related to existing work. This isn't to suggest that the ideas are identical, more that the contribution appears to be incremental.

That being said, I think the analysis is beneficial to the community so give it a weak accept.

---

> ### Author Response · Authors · 2021-11-18
> **Response to gVtv**
>
> Thank you for your kind review! We hope to address your questions and concerns below.
>
> > To state the obvious, such a network is only really applicable to data where you do not need to consider edge features explicitly
>
> We have updated the manuscript to make this clearer. Although we can handle heterogeneous graphs, we leave handling edge features as future work. In the future our approach could be generalized to line graphs [1], and hence incorporate edge features. Similarly, our approach could be generalized to SIN [2]. In principle both these extensions would help with handling edge information.
>
> > The text seems to imply that the computational efficiencies derive from the use of sparse matrix multiplies. But, many new accelerators do not support sparse matmuls well at all, instead are very strong on dense compute. This distinction should be clearer in the text - this is a GPU optimised GNN, not an accelerator optimised GNN (perhaps a more parallel GNN would count for something more general purpose).
>
> It’s true that commercially available accelerators do not offer great support for sparse matrices yet. However, we would argue that sparsity support is a very popular area for research, and we will start to see these developments arrive in the commercial space soon. In particular, we are seeing substantial interest in supporting unstructured sparsity in hardware (i.e. SpMM), and there have been several works dedicated to investigating GNN acceleration at hardware conferences. We have cited a few in the manuscript. Reviewer x8K1 agrees and mentions a few more recent ones in their review.
>
> An important observation is that hardware acceleration for very expressive GNNs is rare in the literature. Most works focus on supporting SpMM-based models, which EGC falls into, and can therefore be run on existing proposed accelerators.
>
> > Please remove the phrase state of the art.
>
> We have qualified our achievements more carefully. (We have updated the abstract in the paper, but we cannot do it on the submission for now. We will handle it for camera ready)
>
> > it is very related to concepts such as attention
>
> Thank you for raising this point. We remain reluctant about making a connection between what we propose and attention, since there is no notion of pairwise similarity in our approach, which we would argue is the defining aspect of attention mechanisms. We believe that our approach is a “dynamic convolutional” model rather than an attentional model.
>
> We do not hide the fact that there are good parallels to our approach in the Transformer literature. In particular, we note that dynamic convolutional approaches such as those proposed by Wu et al [3] also achieve excellent results. We will look to expand this discussion in the appendix for the camera ready version.
>
> > perhaps explain why this model does not perform as well as MolHIV.
>
> This is a really interesting question. MolHIV is a tricky dataset, and there is often a large difference between validation performance and test performance; it is also very easy to overfit. We suspect that the main reason for relative underperformance on MolHIV is the way the data is split: the OGB benchmark splits based on structural properties, whereas ZINC uses a random split. We note that on the OGB leaderboards that methods incorporating more structural information (e.g. CIN/SIN) achieve a far smaller gap between validation and test performance.
>
> We did not spend much time tuning the model parameters for MolHIV, and instead used the setup provided by PNA (which is likely optimized for their model).
>
> > You cite "Training graph neural networks with 1000 layers" which I believe scales O(V), and I think achieves better results on OGBN-ARXIV.
>
> Our work and this work are orthogonal, and can be usefully combined. The focus of the referenced work is to develop strategies to build (very) deep GNN models by preventing the memory consumption from growing with depth. They are not trying to address the cost of a single message passing layer directly -- hence they cannot be usefully described as O(V) or O(E).
>
> With respect to the results on OGB-Arxiv, we would like to make clear that we restrict the parameter count of all our models to 100k. The results in the referenced paper use far more parameters (262k for their smallest model), making it hard to directly compare. We note that their results for GCN and GraphSAGE are also far lower than ours, despite them using more parameters: we have tried to be as fair as possible to the baselines, and they have been tuned extensively.
>
> [1] “Supervised Community Detection with Line Graph Neural Networks” by Chen et al.
> [2] “Weisfeiler and Lehman Go Topological: Message Passing Simplicial Networks” by Bodnar et al.
> [3] “Pay Less Attention with Lightweight and Dynamic Convolutions” by Wu et al.

---

> > ### Comment · Reviewer_gVtv · 2021-11-19
> > **Response to rebuttal**
> >
> > Thank you for your responses, I am mostly convinced by what you report.
> >
> > "It’s true that commercially available accelerators do not offer great support for sparse matrices yet. However, we would argue that sparsity support is a very popular area for research, and we will start to see these developments arrive in the commercial space soon. In particular, we are seeing substantial interest in supporting unstructured sparsity in hardware (i.e. SpMM), and there have been several works dedicated to investigating GNN acceleration at hardware conferences. We have cited a few in the manuscript. Reviewer x8K1 agrees and mentions a few more recent ones in their review.
> >
> > An important observation is that hardware acceleration for very expressive GNNs is rare in the literature. Most works focus on supporting SpMM-based models, which EGC falls into, and can therefore be run on existing proposed accelerators."
> >
> > For what it's worth, I think this is weakest area of your paper. If I understand you correctly, the claim is that *if* we had very good SpMM hardware, and did not need more expressive GNNs, and were not bottlenecked by dense compute (which you could be if you had a GCN with very large MLPs on the node updates), then this model would be very hardware efficient. That might be true, but as you say, doesn't reflect the state of existing commercially available hardware, and is not relevant for many areas of GNN research. This isn't to say your point isn't valid - just that it is limited in scope, and I don't think this limited scope is reflected in your work.
> >
> > Overall, I think this model idea is cool, and the results seems good, so I will keep my score the same. I think a lot of work has gone into evaluating the model, which I appreciate. But, my score will stay the same because of the reason I cite above, which to me limits the potential impact of the paper. Overall, however, the paper should probably be accepted.

---

### Official Review · Reviewer_SZEe · 2021-11-08

**Correctness:** 3
**Technical Novelty And Significance:** 3
**Empirical Novelty And Significance:** 2
**Recommendation:** 3
**Confidence:** 4

**Main Review:**

### Strengths:
1. This paper applied the idea of "basis weights" and thus effectively uses different weight matrices for different nodes.
2. This paper has provided a long section to help interpret the proposed architecture and compared it with GCN, GAT, and PNA.

### Questions:
1. Since it is noted that different aggregators might result in outputs with different means and variances, for the experiments in Table 1, did you normalize the combination weights $w$ or the different aggregated outputs?
2. I am curious how you implemented the "weighting of bases" per node, i.e., multiplication between the combination weighting coefficients $w$ (different for different nodes) and the aggregated outputs corresponding to different basis weight matrices? Is this step before or after the aggregation? Although the theoretical complexity of this step is $O(V)$, I think the choice of actual implementation might significantly affect the practical time efficiency.

### Weaknesses:
1. The major weakness of this paper, in my opinion, is that this paper seems to have misunderstood the real bottleneck to scale-up GNNs, and the proposed architecture may not be suitable for large graphs (i.e., size over a million).
   - The paper has compared with many existing approaches to solve the scalability issues of GNNs on large graphs. However, this paper did not tackle this problem. Firstly, the theoretical complexities of the proposed architecture are still $O(V)$. Thus on a graph with more than a million nodes (which happens in some node classification or link prediction benchmarks), the memory consumption is still too large for conventional GPUs. Secondly, the main experiments (Table 1) are conducted on four graph regression/classification benchmarks and only one node classification benchmark. However, the graphs in the four graph-level benchmarks are small. The experimental exploration and evidence on large graphs are insufficient.
   - For graph sampling approaches, the paper claimed that `We evaluated a variety of sampling approaches and observed that even modest sampling levels, which provide little benefit to memory or latency, cause model performance to decline noticeably.` The corresponding experimental results are shown in Table 8 in the appendix. However, all four benchmarks used in Table 8 are graph-level tasks (graph regression or classification), where the sizes of input graphs are small. For example, ZINC and MolHIV are molecule graphs, and CIFAR consists of super-pixel sampled graphs whose sizes are at most 150 (Dwivedi et al., 2020). Applying sampling strategies to these small input graphs is not reasonable, and it is not strange to observe significant performance degradation. Actually, the performance of some sampling algorithms on relatively large graphs for node classification is strong, e.g., GraphSAINT (SAGE aggregator) can outperform the "Full-batch" GraphSAGE on *ogbn-products* (see [leaderboard](https://ogb.stanford.edu/docs/leader_nodeprop/)). The quoted sentence above is not appropriate.
   - To sum up, I think the architecture proposed in the paper does not solve the scalability problem on large graphs. Thus the authors should make this point clearer in the related work and experiment sections.
2. Another potential problem, in my opinion, is the spectral interpretation in Section 4.2. The paper claims that `Our approach corresponds to learning multiple filters and computing a linear combination of the resulting filters with weights depending on the attributes of each node locally.` The paper compared two formulas: (1) the ideal linear combination of filters: $y=\sum_{b=1}^Bw_b\odot g_{\theta_b}(\mathbf{L})\mathbf{X}$; and (2) the filtering of proposed model (correspond to EGC-S, Eq.(2)) $y=\sum_{b=1}^Bw_b\odot (\tilde{D}^{-\frac12}\tilde{A}\tilde{D}^{-\frac12})\mathbf{X}\Theta_b$.  However, I think the latter formula is not a combination of multiple filters. There is only one filter $(\tilde{D}^{-\frac12}\tilde{A}\tilde{D}^{-\frac12})$ (which is left multiplied to $\mathbf{X}$) and the weight matrix $\Theta_b$ right multiplied to $\mathbf{X}$ should not change the relative magnitude of signals of different frequencies. It is not the same if changing $g(\mathbf{L})$ in the former formula, where the matrix left multiplied to $\mathbf{X}$ changes.

**Summary Of The Paper:**

This paper claimed they designed a new GNN architecture that achieves state-of-the-art performance with lower memory consumption and latency. More specifically, the proposed model uses memory proportional to the number of vertices in the graph $O(V)$, in contrast to competing methods which require memory proportional to the number of edges $O(E)$. The paper claimed that the new architecture enabled each vertex to have its own weight matrix, thus following a novel adaptive filtering approach. The experiments found that the proposed efficient model could achieve higher accuracy than competing approaches across six large and varied datasets against strong baselines. Moreover, the experiments demonstrated that the proposed method achieves lower latency and memory consumption for the same accuracy compared to competing approaches.

**Summary Of The Review:**

This paper proposed a novel architecture to linearly combine the aggregated outputs using different weight matrices for each node. The model is memory efficient since the memory consumption is $O(V)$ instead of $O(E)$. However, the proposed method did not solve the scalability problem of GNNs when applied to considerably large graphs. And it is unclear when the provided memory efficiency is necessary, and thus the proposed method is the first choice. In terms of performance, the improvements compared with PNA are also not very convincing. The significance of this paper might be limited if the efficiency/performance improvements are marginal. I would encourage the author to explore the theoretical understanding of the proposed architecture further. Currently, section 4 is well-written, but the conclusions are limited and may have some flaws. In general, I could not recommend the current manuscript for acceptance.

---

> ### Author Response · Authors · 2021-11-18
> **Response to SZEe**
>
> Thank you for your review. We will first address your questions, before looking to address your concerns.
>
> In response to your questions:
>
> 1. We do not normalise $w$. Indeed,  this likely impacts EGC-M model performance (although it is still the best performing model), but we also note that inserting a normalization layer prevents usage of optimizations such as aggregator fusion.
> 2. This step is after aggregation, as indicated in equation 4 in the paper. You can also see this implemented in our codebase -- we provide both layers.py and opt_layers.py to allow you to see both a “simple” and a more optimized implementation of the model.
>
> We now proceed to address your concerns.
>
> > The paper has compared with many existing approaches to solve the scalability issues of GNNs on large graphs
>
> Thank you for raising this; we did not want to imply that we “solve” the issue -- we are presenting one orthogonal improvement that can be combined with other approaches. We have updated the manuscript to make this clearer. The major claim of this work is that we can achieve similar performance to the O(E) anisotropic mechanisms that are prevalent in the GNN literature in just O(V) memory.
>
> Following your suggestion we have emphasized that you can combine methods in the background section.
>
> > Thus on a graph with more than a million nodes (which happens in some node classification or link prediction benchmarks), the memory consumption is still too large for conventional GPUs
>
> We provide (full-batch) results on OGB-MAG, which has 2M nodes and 21M edges; we also note that this is a heterogeneous graph. We kindly ask that you reconsider this claim.
>
> > The experimental exploration and evidence on large graphs are insufficient.
>
> Please refer to our results on OGB-MAG, a large graph dataset. We show on this dataset that we perform strongly on large graphs and heterogeneous data.
>
> > For graph sampling approaches, the paper claimed that We evaluated a variety of sampling approaches and observed that even modest sampling levels, which provide little benefit to memory or latency, cause model performance to decline noticeably. The corresponding experimental results are shown in Table 8 in the appendix. However, all four benchmarks used in Table 8 are graph-level tasks (graph regression or classification), where the sizes of input graphs are small [...] Applying sampling strategies to these small input graphs is not reasonable, and it is not strange to observe significant performance degradation
>
> It is important to note the sentence we included immediately before the sentence you quoted: “Sampling methods are often ineffective when applied to many problems which involve **model generalization to unseen graphs**---a common use-case for GNNs.” [emphasis ours]
>
> We explicitly agree with you that applying sampling to small (unseen) graphs is not a reasonable application. However, we are not allowed to make these claims without evidence -- hence why we have provided numeric results.
>
> >  the performance of some sampling algorithms on relatively large graphs for node classification is strong, e.g., GraphSAINT (SAGE aggregator) can outperform the "Full-batch" GraphSAGE
>
> We agree here, but we did not make this claim. Our comment was scoped exclusively to problems where we need to generalize to unseen graphs.
>
> > However, I think the latter formula is not a combination of multiple filters.
>
> The filter used on the convolution is not just the normalised Laplacian $\tilde{D}^{-\frac{1}{2}} \tilde{A} \tilde{D}^{-\frac{1}{2}}$ but the product of normalised Laplacian and a weight. This comes from the fact that the filter is approximated using a polynomial of degree 1; the weights correspond to the polynomial coefficients (cf. Eq. (4) in “Semi-Supervised Classification with Graph Convolutional Networks” by Kipf and Welling 2017). The weights hence determine how strongly the convolution acts as a low-pass filter (see also “Simplifying Graph Convolutional Networks” by Wu et al. 2019).
>
> > And it is unclear when the provided memory efficiency is necessary, and thus the proposed method is the first choice
>
> The scenario we are most concerned with is to do with models where we need to generalize to unseen graphs at inference time. This situation comes up commonly: e.g. code analysis / compilers or point cloud applications [1] are both good examples. Sampling methods cannot help reduce inference time here (if anything, they increase it overall). We discuss this at length in the background section.
>
> > In terms of performance, the improvements compared with PNA are also not very convincing
>
> It is unclear how you arrive at this conclusion. We kindly ask that you expand this comment. We achieve better results than PNA with lower memory consumption and latency. We argue that achieving both simultaneously is an impressive achievement.
>
> [1] “Towards Efficient Point Cloud Graph Neural Networks Through Architectural Simplification” by Tailor et al.

---

### Author Response · Authors · 2021-11-18
**Response to all reviewers**

We thank all the reviewers for their time. We really appreciate your thoughts, and we firmly believe that they have helped improve the paper. We will update the pdf shortly.

Please do let us know if you have any more questions!

---

### Decision · Program_Chairs · 2022-01-20

**Decision:**

Accept (Poster)

**Comment:**

The manuscript develops a new and simple graph neural network architecture. The proposal make use of only O(V) (number of vertices) rather than O(E) (number of edges, meaning that it may be useful for scaling to larger problems. The didactic figures are especially clear, and as is shown in Fig 1 the proposed architecture passes messages based only on the source vertex rather than based on source and target. This challenges common ideas in the field that passed messages ought to reflect a function of both source and target. In spite of this introduced simplification, the architecture performed better than or as well as a set of strong baselines on a set of 6 datasets. The manuscript also examines latency and memory consumption, showing that the methods comes out favourably in this regard.
One of the reviewers worries that the paper does not directly provide a solution to scaling network training to very large graphs; they note that several of the datasets that are examined do not contain large graphs. This is true, but the paper does not overclaim in this regard, and I agree with the majority of reviewers that the manuscript is worth publishing on the basis of having developed a well-performing approach that challenges the accepted assumptions in the field. While it may not be a direct solution, the counterintuitive results may help point the direction toward development of simple, effective approaches that do scale up.